# Sensorizing a Beehive: A Study on Potential Embedded Solutions for Internal Contactless Monitoring of Bees Activity

**DOI:** 10.3390/s24165270

**Published:** 2024-08-14

**Authors:** Massimiliano Micheli, Giulia Papa, Ilaria Negri, Matteo Lancini, Cristina Nuzzi, Simone Pasinetti

**Affiliations:** 1Department of Mechanical and Industrial Engineering (DIMI), University of Brescia, Via Branze 38, 25128 Brescia, Italy; massimiliano.micheli@unibs.it (M.M.); simone.pasinetti@unibs.it (S.P.); 2Department of Sustainable Crop Production (DI.PRO.VE.S.), Catholic University of the Sacred Heart, Via E. Parmense 84, 29122 Piacenza, Italy; giulia.papa@unicatt.it (G.P.); ilaria.negri@unicatt.it (I.N.); 3Department of Medical and Surgical Specialties, Radiological Sciences, and Public Health (DSMC), University of Brescia, Viale Europa 11, 25128 Brescia, Italy

**Keywords:** measurement science, hardware instrumentation, honeybee monitoring, precision agriculture, *Varroa destructor*, embedded measurements, object detection

## Abstract

Winter is the season of main concern for beekeepers since the temperature, humidity, and potential infection from mites and other diseases may lead the colony to death. As a consequence, beekeepers perform invasive checks on the colonies, exposing them to further harm. This paper proposes a novel design of an instrumented beehive involving color cameras placed inside the beehive and at the bottom of it, paving the way for new frontiers in beehive monitoring. The overall acquisition system is described focusing on design choices towards an effective solution for internal, contactless, and stress-free beehive monitoring. To validate our approach, we conducted an experimental campaign in 2023 and analyzed the collected images with YOLOv8 to understand if the proposed solution can be useful for beekeepers and what kind of information can be derived from this kind of monitoring, including the presence of *Varroa destructor* mites inside the beehive. We experimentally found that the observation point inside the beehive is the most challenging due to the frequent movements of the bees and the difficulties related to obtaining in-focus images. However, from these images, it is possible to find *Varroa destructor* mites. On the other hand, the observation point at the bottom of the beehive showed great potential for understanding the overall activity of the colony.

## 1. Introduction

Despite being a relatively young sector, Italian honey and bee wax production is extremely flourishing. According to the 2024 report of iCRIBIS [1], there are more than 1.6 million beehives located mostly in the northwest (32.1% compared to 19.1% in the northeast, 18.2% in the center, 17.2% in the south, and 13.4% on the two isles of Sicily and Sardinia). Of these beehives, 79% are managed by professional beekeepers running mostly micro companies (fewer than 10 people), and 25.6% of them are managed by young professionals. Micro companies typically lack the funding to invest in technological advances; thus, they are forced to rely on traditional methods instead of digital solutions due to the costs of the hardware and the lack of experience in operating it. However, innovation in this sector can still be achieved by leveraging simple and low-cost solutions [2].

The European honey bee *Apis mellifera* is a semi-domesticated species that can populate artificial beehives (such as the common Dadant beehive [3]) for commercial purposes [4]. Bee colonies are subject to human selection but are not restricted in their foraging since they can freely access the surrounding landscape, thus requiring the beekeepers to place the beehives in a protected environment to ensure a good quality product. Unfortunately, external stressors such as weather conditions, pesticide applications, habitat quality, and potential infections due to viruses and pests can affect a colony’s health and determine its mortality [5]. In particular, winter plays a huge role in the survival of a colony because of the temperature drop, which could cause stress to the colony and its queen [6,7], and the potential infection of mites during fall that spreads inside the colony undisturbed during winter [8]. Among the potential parasites, the most dangerous one is *Varroa destructor*, a red and round mite of 2 mm diameter that attacks and feeds on the bees [9,10]. If the infestation is not treated in time, the mite population grows during winter, and genomic diseases are passed on to newborn bees, leading to the colony’s death [11,12].

As a result, to ensure the colony overwintering, beekeepers have to check on the colony’s health from time to time, provide enough food to sustain the colony’s well-being, and eventually perform treatments to remove mites and other parasites. These are, however, invasive procedures because by removing the protective cover of the beehive, the bees are exposed to frigid temperatures, and their activity is disturbed, sometimes in irreparable ways. This is why the research community tried to come up with solutions to monitor the overall health and status of the colony using sensors and digital devices. In ref. [13], the authors discussed an approach based on image analysis to detect the presence of *Varroa destructor* mites on the bees and an audio-based method to analyze the sounds produced by the colony and understand its health status. However, both methods were only discussed as preliminary and using laboratory data such as good-quality images and public audio datasets. Audio-based methods were also studied in refs. [14,15], in conjunction with other sensors such as internal temperature and humidity for better prediction results. Other analyses were conducted involving weight sensors [16] to periodically check the production and health status, and colony anomalies based on swarms prediction [17]. Low-cost multi-sensor approaches such as refs. [18,19,20] propose integrated solutions with the beehive and Internet of Things communication protocols for data transfer, analysis, and storage. By combining different sources of information, it is possible to obtain a comprehensive analysis of the colony status, and this is why such solutions are recently preferable [2].

However, these works do not include vision systems to monitor the colony’s status inside the beehive despite being a successful inspection method as highlighted by refs. [21,22], where images of infected bees were analyzed in an automated way to detect the presence of the parasite. The only solution that does actually include a vision system inside the beehive (hence, it runs on real operative conditions instead of laboratory ones) is presented in ref. [23]. This work involved an infrared camera mounted inside the beehive’s box; however, no experimental data nor feasibility information were reported apart from the prototype design.

This study presents a camera-based monitoring system designed for observing bees throughout the colder seasons, which, to the best of our knowledge, has never been proposed before. In Section 2, particular attention is paid to the technical strategies for simultaneously monitoring beehives from different observation points without harming them, evaluating the feasibility of each developed solution compared to our initial design presented in ref. [24]. Preliminary data collected during experimental campaigns are used to validate the proposed instrumentation design and analyze if the observation points of the commercial beehive can be informative enough to draw relevant conclusions about the beehive’s status as discussed in Section 3. Results of the experimental campaigns conducted in 2023 are presented in Section 4 and, despite not being the focus of this work, we include a small discussion about the potential detection of *Varroa destructor* inside the beehive with our solution. Considering the promising results obtained from the experiments, we are positive that our sensor-based beehive design can pave the way for new frontiers in monitoring paradigms for beekeepers.

## 2. Materials

The measurement system has been developed considering the technical constraints and design criteria necessary to respect the daily activities of the beehive during winter:**Space constraints:** The structure of a beehive is very compact, with an intra-frame distance of about 15 mm (typically 10 to 12 frames are used, thus occupying all the hive’s space). Inspection points like the top and bottom of the hive also have limited space. As a result, the developed monitoring device should be small enough to fit the beehives.**Non-invasiveness:** Since bees can be easily disturbed by external devices, light, intense odors, or heat, the developed device should be as least invasive as possible, avoiding interference with the hive’s activities and preventing constrained pathways, both outgoing and incoming from the hive (such as channels as recently proposed by the scientific literature [25]).**Low cost:** The selection of potential hardware for internal monitoring of the beehive should also consider costs alongside technical aspects, in anticipation of potential larger-scale prototypes. While a set-up with industrial cameras and lighting would offer higher data quality, robustness, and speed of implementation, it would be too expensive to be used by beekeepers in practice.**Lighting conditions:** Observing bees within their hive during winter presents a significant technical challenge. Limited daylight hours, the number of bees inside the hive, and the near absence of light within the hive itself make it crucial to develop customized solutions to ensure sufficient brightness to obtain quality images while keeping the invasiveness of the illumination system to a minimum.**Flexibility:** During winter, the beehive forms a cluster, known as the *glomerulus*, a mass of bees that sequentially maintains a stable temperature inside the hive to endure the harsh winter temperatures. The *glomerulus* shifts to different locations in the hive for various reasons, primarily to access nectar stores necessary for temperature maintenance. Therefore, a flexible set-up should be able to track the movement of the *glomerulus* and ensure continuous observation of the bees.

Before establishing the measurement set-up, a team of expert entomologists from the University of the Sacred Heart conducted an assessment of the Dadant model beehive [3] used in this work. They evaluated the internal dimensions of the beehive, identified key points where bees perform their primary activities, and assessed the overall functionalities of the beehive. This evaluation aimed to pinpoint measurement locations and strategically position the hardware and lighting system, ensuring alignment with the above-mentioned criteria. Therefore, two points of interest were identified, where the bee colony primarily engages in its activities, taking into account the natural behavior of bees during winter: (i) in-between frames, close to the brood (“Frame”), and (ii) looking up from the bottom section of the beehive, where bees move to change frames and the eventual dead bees fall to be collected by the bottom plate (“Bottom”). Notably, entrance monitoring was excluded due to minimal to negligible bee activity during winter. Given its proximity to the bees inside the beehive, the “Frame” inspection point could potentially serve as a sentinel of pest infections such as *Varroa destructor*, which is a parasite affecting young bees [9].

### 2.1. Developed Set-Up

Considering the technical constraints described in Section 2, the experimental beehive was instrumented considering the two points of interest: (i) inside the beehive (“Frame” Camera), and (ii) on the bottom of the beehive (“Bottom” Camera). The electrical components to operate the hardware were placed in the *honey super* to provide isolation from rainfalls.

The camera model was the same for each point of interest, consisting of an RGB camera manufactured by The Imaging Source (NC, USA) model DFK ECU010-M12, with a pixel resolution of 1280×720 px, pixel size 3 × 3 µm, a frame rate of 30 frames per second (fps), and overall dimensions of 22×13×60 mm. This size is perfect for the small available space in the beehive. A novel contribution proposed in this work, following the idea described in our previous work [24], is the integration of the “Frame” Camera within a comb. This innovation results in an instrumented frame capable of observing young bees predominantly engaged in cell cleaning and brood nourishment activities. Given their close interaction with the brood and potential exposure to infections, this arrangement was deemed fundamental by the expert entomologists. Moreover, the camera and all-encompassing lighting have been seamlessly integrated into a single device, positioned at the center of a comb, allowing the monitoring of the comb adjacent to the instrumented one. As a result, while the “Bottom” Camera was equipped with fixed optics with a focal length of 8 mm, the “Frame” Camera utilized liquid lenses, model Caspian M12-316-26 manufactured by Corning Varioptic (Glendale, AZ, USA), mounting a built-in Arctic 316 liquid lens, a special type of optics that can change the optical focal length dynamically according to the external tension applied. This technology allows obtaining clear and sharp images of the inside of the beehive due to the reduced space between the frames [26]. The minimum focal length of this model is 2.6 mm up to infinity, with an angular field of view of $160^{\circ }$, an iris aperture ranging from 2.5 to 10 mm, 4 mm to infinity focus range, and optical power that spans from −15 diopters to +38 diopters, resulting in a 53 diopters dynamic range, depending on the supply voltage. The device’s response time is less than 15 ms after a new tension is provided to change the focal length, and the overall power consumption is 0.1 mW. The device is compatible with up to 1/2.5" image sensors and M12 ×0.5 mounts (S-Mount), as well as off-the-shelf FPC connectors. They are actuated by the control board HV892 manufactured by Supertex Inc. (Sunnyvale, CA, USA) operating in the voltage range of 25 to 60 VDC. The two communicate via the I2C protocol using a dedicated software driver. The cameras are connected to a single Raspberry Pi 4 (Cambridge, England) via a USB interface that manages the custom-made acquisition software and data storage.

It is worth noting that in the case of the “Frame” inspection point, the bees do not move on a single plane. Instead, they transition from one comb to the next, leading to random changes in their distance from the camera. To address this behavior and maximize the number of in-focus images, it is necessary to adjust the focal distance of the liquid lenses instead of choosing a fixed focal length. Auto-focus technology could help in this regard; however, the response times of the liquid lenses are longer than the bees’ dynamics, meaning that the adjusted focal length becomes effective after the bees move. As a workaround, we decide to continuously actuate the liquid lenses using a sinusoidal law at a frequency of 1 Hz in their voltage range (sweep). In this way, a portion of the images will surely be in focus regardless of the bees’ movements and positions. A graphical example of what happens is shown in Figure 1. The sinusoidal law corresponds to the actuation voltage generated by the control board and sent to the liquid lenses to modify the focal length. Hence, for each portion of the sine wave, we can obtain different images according to the focal length. The example shows real data taken from a single sweep, highlighting that most of the images taken are not in focus. Moreover, for each sweep, the in-focus portions change position according to the bees’ movements, preventing calibration. The selection of in-focus images will be described in Section 3.1.

Since the inside of the beehive is dark, a custom-made red LED ring was designed to provide minimum illumination without harming the insects, positioned around the optics of the “Frame” Camera. Three strips of red LEDs were positioned inside the beehive as well to improve the general illumination and benefit images taken from the “Bottom” Camera. Since this camera was placed at the bottom of the beehive, the metallic grid partially occluded its view. Therefore, the LED strips were positioned vertically on both inner hive walls in correspondence with the central frames to minimize reflections resulting from the metal grid. All the LEDs of the illumination system were actuated by the Raspberry Pi which sends a PWM to a Mosfet driver operating at 12 VDC, thus modulating the light intensity according to operational needs. To mitigate overexposure effects, the illumination system’s duty cycle of the LED ring was set to 50%, while the duty cycle of the LED strips placed on the hive’s walls was set to 70%. The red light was chosen because it is the least disruptive wavelength to the colony (which remains in the dark within the beehive). Furthermore, red light has been scientifically tested as a positive treatment during winter [27] because it helps increase the bees’ mitochondrial function, protects them against pesticide exposure, and aids in keeping suitable heat inside the colony during adverse weather conditions.

From a technical standpoint, the “Bottom” Camera was mounted using aluminum profiles that allowed it to glide along the observation plane, facilitating manual adjustments at points of interest with higher activity or bee cluster concentration. The instrumented beehive was placed on an elevated support to allow suitable space for the camera and to avoid stagnant water during and after rainfalls.

A scheme of the final experimental set-up and the wiring of components is shown in Figure 2.

### 2.2. Data Acquisition Software

The cameras were connected to a single Raspberry Pi 4, on which the custom-made acquisition software was installed. The main constraint when working with affordable embedded devices is limited computational power, so the data acquisition software should be optimized to guarantee that no data are lost. The developed software was written in Python 3.9 and leverages the multi-process library to (i) acquire frames from each camera and (ii) save all the stored frames. The idea, illustrated in Figure 3, is to create a new sub-process for each camera dealing with the independent acquisition of the frames whenever the corresponding red light illuminator is active (“Acquisition” sub-processes). These sub-processes acquire frames and store them in a queue, exploiting the device’s RAM. At the same time, the “Saving” sub-processes write the frames to disk until the queue is emptied. When the corresponding red light illuminator is powered off, the “Acquisition” sub-processes stop as well, while the “Saving” sub-processes keep working until their task is completed. The procedure is repeated according to the illumination duty cycle.

The procedure is summarized as follows:The main program automatically detects the number of cameras connected (in our case, only 2) and prepares the saving directories. Due to the USB bandwidth limitation and board processing speed, this set-up can manage up to 4 cameras.The main program manages (i) the triggering of the two red light illuminators, positioned around the “Frame” Camera (led ring) and inside the beehive (led strips), at the predefined duty cycle frequency set during configuration (50% for the led ring, 70% for the led strips), and (ii) the activation signal of the liquid lenses through the dedicated driver connected to the Raspberry Pi 4 (as described in Section 2.1).“Acquisition” sub-processes are launched on separate cores for each connected camera. They capture frames at a resolution of 640×480 pixels and a rate of 30 fps. Each frame is temporarily stored on a queue leveraging the board RAM; hence, to limit the memory usage, each acquisition lasts only 60 s.Concurrently, “Saving” sub-processes for each camera are launched. They grab the frames on top of the corresponding queues and save them onto an external disk.

### 2.3. Acquired Data

Our dataset is composed of images taken from the “Frame” Camera and the “Bottom” Camera. To save space, the original image was saved after resizing to 640×480 px. For the “Frame” Camera, we acquired a total of 17,210 images during the last days of April and the first week of May 2023. However, only a portion of them were in focus, thus requiring a pre-processing procedure to extract only the valid images (as detailed in Section 3.1). Data from the “Bottom” Camera were acquired in September 2023 for a total of 1494 images.

## 3. Methods

### 3.1. In-Focus Images Automatic Selection

This method was developed in Mathworks MATLAB 2023b (Natick, MA, USA) and applied only to the data acquired from the “Frame” Camera since the optimal focus of the liquid lenses was unknown and varied according to the movements of the bees in the FoV.

As described in ref. [28], a variety of *focus measure operators* can be computed on an image to compute its overall *focus measure*. Focus measure operators are obtained pixel by pixel, and the resulting value of the *focus measure* of an image depends on the image noise level, contrast, saturation, and window size. As a result, by changing the acquisition conditions (e.g., the camera, the lenses, and the illumination), the values of the *focus measure* considered to separate in-focus images from unfocused ones may change, thereby requiring a specific calibration step to identify their values.

Among the plethora of *focus measure operators*, we experimentally selected only the ones that showed distinct differences between in-focus and unfocused images in our dataset. These are Gaussian derivative (GDER), Gray-level local variance (GLLV), Steerable filters (SFIL), Tenengrad (TENG), and Tenengrad variance (TENV). For their mathematical and MATLAB implementations, see ref. [28] and ref. [29] respectively.

We computed the values of the 5 *focus measure operators* for each image in our dataset and stored them in a table. We manually labeled a subset of 200 images as in focus or unfocused, and plotted their distribution, highlighting two distinct clusters. As a result, we found that in-focus images have GDER ≥26, GLLV ≥0.5, SFIL ≥ 1.1 · 10^18^, TENG ≥400, and TENV $\geq 480\cdot 10^{3}$. These values were used as thresholds to identify in-focus images automatically. As a result, from the original 17,210 images, we obtained 2057 images (12% of the original dataset). Since we aim to use these images to perform transfer learning on a pre-trained object detector, this number is sufficient to reach a satisfactory level of accuracy (for transfer learning, even hundreds of images are typically enough [30,31]). It is worth noting that among the in-focus images, bees were not always present or fully visible, leading to a two-class detection problem as described in Section 3.2.1.

To visualize how the in-focus images are expressed in a five-dimension space, we used a dimensionality reduction algorithm named t-SNE (t-distributed Stochastic Neighbor Embedding) [32], which is a non-linear alternative to PCA (Principal Component Analysis). We reduced the visualization space to 2D, highlighting in green the instances of “in-focus” images selected by our thresholding algorithm (Figure 4a). Apart from a few belonging to a separate cluster (e.g., containing image features different from the other cluster, such as a cluttered view of several bees on top of each other or empty images depicting the honeycombs only), the main cluster is well separated from the rest (black points). For better visualization, we also obtained the 3D visualization of the same reduced space in Figure 4b. In this case, the in-focus cluster (green) is almost completely separated from the rest, highlighting the validity of our custom thresholding procedure.

### 3.2. Bees Detection

The model adopted for the automatic detection of the bees was the state-of-the-art open-source object detector YOLO v8.2 of Ultralytics [33]. The model used was the pre-trained version from the Ultralytics library, allowing us to perform transfer learning instead of training from scratch. Transfer learning is a common procedure adopted by the research community to exploit models trained on large datasets with a general task (e.g., recognize up to 1000 different classes) and “refine” them to work on a very specific subset of classes by retraining only a few levels of the network. Consequently, only a few hundred images are typically sufficient to reach satisfactory results [30,31].

We trained the models on Google Colab [34], a web service based on Jupyter Notebook that provides remote access to computing machines equipped with GPUs. We used the standard plan, which provides a machine with RAM of 12 GB, disk space of 78 GB, and a GPU of 15 GB. The training code was developed in Python 3. Training parameters were the same for the two models: epochs 300, patience 50, image batch 16, intersection over union 0.7, momentum 0.937, and weight decay 0.0005. Images were resized to 640×640 px.

All data were manually labeled by an expert for the object detection task, thus identifying a number of rectangles (*bounding boxes*) corresponding to the bees inside the image. Each bounding box is defined in the image plane (in pixels) with four parameters: the coordinates of the box center $C_{X}$ and $C_{Y}$ and the box size *W* (width) and *H* (height). To use these four parameters in the model, their original values must be normalized over the image width $I_{W}$ ($C_{X}$ and *W*) and height $I_{H}$ ($C_{Y}$ and *H*) respectively, obtaining values in the range [0, 1]. An example is depicted in Figure 5, highlighting the image coordinate system origin centered on the top-left corner.

We wish to stress that the analysis conducted with the YOLO detector has the sole purpose of studying the behavior of the bees and understanding if the proposed instrumentation set-up could provide insightful information for beekeepers. As a result, the detector of choice is merely a tool; it could even be another one among the plethora of state-of-the-art detectors, and we do not aim to prove its ability to detect bees.

#### 3.2.1. “Frame Detector” Dataset

The 2057 in-focus images of the “Frame” Camera were divided as follows: 1440 images were used for training (70% of the total), 411 for validation (20% of the total), and 206 for testing (10% of the total). This is a standard subdivision adopted by the research community dealing with artificial intelligence models (refer to ref. [31] for further explanations). We used two classes: “bee”, referring to an in-focus and fully visible bee, and “blurred_bee”, corresponding to unfocused bees or portions of them (e.g., the head or the back). This classification was necessary since some bees were captured while moving; hence, they were blurred or only partially visible. However, since the objective of the “Frame detector” is to monitor the bees’ activity, counting how many bees passed in front of the camera is useful information that any detector can obtain. In this sense, counting blurred and occluded bees is also necessary. Examples taken from the dataset are shown in Figure 6.

#### 3.2.2. “Bottom Detector” Dataset

The dataset of 1494 images taken from the “Bottom” Camera was divided into training (1044 images, 70% of the total), validation (303 images, 20% of the total), and test (147 images, 10% of the total) sub-datasets. We used two classes: “bee”, referring to a perfectly visible bee, and “occluded_bee”, corresponding to a bee partially occluded by the bottom grid of the honeycomb or appearing at the edges of the FoV. Examples taken from the dataset are shown in Figure 7.

#### 3.2.3. Detection Metrics

Since experts manually labeled the bees depicted in the images of both datasets, a first analysis can be conducted on the labels themselves. This helps us understand in which area of the image the bees appear more frequently and, hence, if the point of interest is meaningful or not, and if there are visual disturbances that limit its efficacy. To this end, we computed the correlogram of the labels for both datasets (see [35] for technical details). It is worth noting that, to ensure that no labeling mistakes (e.g., wrong class assignment and misplaced bounding box) were made during the labeling process, another expert not previously involved in the process checked the labels of both the “Frame” and “Bottom” datasets. A correlogram is a group of 2D histograms and heatmaps showing, for each axis combination, the distribution of the bounding boxes’ coordinates ($C_{X}$, $C_{Y}$, *W*, and *H*) in the range [0, 1] (normalized coordinates space). It is drawn as a triangular matrix of graphs of 4 rows × 4 columns, only showing the diagonal and the bottom portion of the matrix. For convenience, we will refer to individual graphs considering their location as (row, column) pairs, where the first row is the top one and the first column is the left one. Darker colors of the heatmaps represent more frequent occurrences in the dataset.

Secondly, after training the two detectors “Frame detector” and “Bottom detector” on the corresponding datasets, we analyze their performance according to a set of common parameters used by the research community: precision, recall, and mean average precision (mAP) [33,36]. Precision refers to the number of true positives (TPs) among all the positive predictions (considering also the false positives, FPs), while recall refers to the number of TPs among all the predictions (considering also the false negatives, FNs). Hence, if precision measures the model’s capability to avoid FPs, recall measures the model’s ability to detect positive instances of a specific class. Both values are in the range [0, 1] and represent proportions. If precision and recall are metrics typically used for any AI model, mAP only refers to object detectors. This metric was created specifically to understand if the positioning of the bounding box (or mask) is correct by exploiting another metric called Intersection over Union (IoU), which specifies how much of the predicted bounding box area overlaps the ground-truth box area. As a result, object detectors FPs can be produced in two cases: (i) when a box is correctly overlapping the ground-truth but the class is wrong, or (ii) when the class of the predicted box is correct but the IoU is below a threshold, meaning that the predicted box’s area is mostly outside the ground-truth box area. This also extends the general definition of precision and recall since the conditions to obtain TPs, FPs, TNs, and FNs change according to the IoU threshold specified. The average precision (AP) of a model is therefore calculated by first obtaining the precision-recall curve and then calculating the area under it. However, since the curve may have a non-monotonous pattern, it is typically smoothed considering that at each recall value $X_{i}$, the corresponding precision $Y_{i}$ is replaced by the maximum precision value located on the right of that particular point ($X_{N}$, $Y_{max}$), considering N>i and that $X_{N}$ corresponds to $Y_{max}$ which is a local maximum. Hence, at each $X_{i}$, the corresponding $Y_{i}$ is substituted with the maximum value $Y_{max}$. This smoothing principle takes into consideration the eventual spikes in the curve, and the result is a curve that has a monotonically decreasing step shape, which is easier to interpolate to obtain AP. Finally, mAP is obtained by finding the AP of each class of the model and then averaging it for the total number of classes. Therefore, mAP is the metric to look at when training multi-class object detectors. Several mAP can be obtained according to the IoU threshold considered to calculate AP; in this work, we will analyze mAP-50 (AP calculated at IoU = 50%, considered an “easy” threshold, hence allowing a lot of incorrect positioning of the boxes) and mAP-50-95 (AP calculated considering varying IoU thresholds ranging from 50% to 95%, thus giving a comprehensive view of the model performance at different levels of detection difficulties), which are the standard outputs of YOLO detectors.

The final metric we will consider is the confusion matrix [37] computed on the test sub-dataset of each model (“Frame detector” and “Bottom detector”), also comprehending the row-wise and column-wise summaries representing the True Positive Rate (TPR), False Negative Rate (FNR), Positive Predicted Values (PPVs), and False Discovery Rates (FDRs). This is a powerful tool that helps us understand if the model correctly detects the bees and if it is able to distinguish between the two classes (“bee” against “blurred_bee” or “occluded_bee”) and the background (corresponding to a “no detection” class or a “true negative” instance). It provides a summary of the model’s performance at a glance thanks to the heatmap representation and row-wise and column-wise summary that represent the True Positive Rate (TPR), False Negative Rate (FNR), Positive Predicted Values (PPVs), and False Discovery Rates (FDRs). TPR and FNR represent the proportion of correctly and incorrectly classified observations per true class, respectively. These two values are obtained as a row summary of the confusion matrix. PPV and FDR represent the proportion of correctly and incorrectly classified observations per predicted class, respectively, obtained as the column-summary of the confusion matrix. They are a useful metric to understand the behavior of the model with respect to false positive predictions.

## 4. Results

### 4.1. “Frame Detector” Model Results

Looking at Figure 8, which shows the correlogram plot of all the ground-truth labels of the dataset, we can observe the following:The boxes’ center is located everywhere in the image, mostly in the middle and in the bottom-left corner (graph at row 2, column 1). This is also highlighted by the two histograms at (row 1, column 1) and (row 2, column 2).The relationship between $C_{X}$ and *W* (row 3, column 1) and $C_{Y}$ and *H* (row 4, column 2) similarly depicts an arrow-like shape. This shape suggests that boxes with the $C_{X}$ or $C_{Y}$ coordinate around 0.5 (middle of the image) have bigger dimensions, while at the edges of the frame, the bounding box size is smaller. At the same time, by looking at the colors of the squares in the graphs (darker means more occurrences), we can deduce that boxes at the frame’s edges frequently have W>H, while at the center, it is more common to have W<H.Looking at the graph at (row 3, column 2), for $C_{Y}$ coordinates corresponding to the center and bottom parts of the frame, the bounding boxes have *W* values ranging from 0.5 to 0.25. Considering that the graph shows all the combinations of $C_{Y}$ and *W* values, this may suggest that the bees identified in those image portions are usually not fully visible (hence, the bounding box is thin) or laterally rotated. This claim is also backed up by the other graph in (row 4, column 1) showing the relationship between $C_{X}$ and *H*.Considering the graph at (row 4, column 3), it is evident that the bounding boxes are typically big, with *H* close to 1 and *W* close to 0.5 (probably bees that appear in the camera’s FoV from the bottom moving upwards or vice versa), or very shallow, with *H* close to 0.5 and *W* close to 0.2 (probably bees appearing in the camera’s FoV but not fully visible). This correlation is also suggested by the two histograms at (row 3, column 3) and (row 4, column 4).

The model’s performance is shown in Table 1. The metrics described in Section 3.2.3 have been calculated for each sub-dataset (training, validation, and test) and are presented subdivided per class, including a general class “all” that combines the results from all the individual classes. Looking at the results, it is evident that class “bee” does not achieve satisfactory results and this may be due to the labeling. The labeling guidelines adopted define a “bee” as an in-focus bee that is typically fully visible, while a “blurred_bee” is not only a blurred bee that is fully visible but also portions of the bee, such as the back or the head. Moreover, our dataset includes just a few instances of a “bee” as defined earlier (147 among the 2057 images of the dataset); hence, the higher number of “blurred_bee” instances (3795 among the 2057 images of the dataset) in the whole dataset could have led the model to wrong classifications of the occurrences quite often (unbalanced classes problem). In contrast with the low performance of the model on “bee” occurrences, the performance on “blurred_bee” is promising since it surpasses 80% for precision, recall, and mAP-50 on the test sub-dataset. It is worth noting that the results on mAP-50-95 are always below 80% (78% on the training sub-dataset, 64 on the validation sub-dataset, and 67% on the test sub-dataset), indicating that the model does not perform well if we consider an IoU confidence higher than 50% (as described in Section 3.2.3). This situation is also underlined by the confusion matrix in Figure 9, showing that for class “bee”, the TPR is only 60% compared to the FNR at 40% (instances classified as “blurred_bee”). In contrast, for class “blurred_bee”, the TPR is 86.7% with a FNR of just 13.3%. True negative instances (e.g., images with no instances of either “bee” or “blurred_bee”) are always classified wrongly, and this could be due to the presence of the honeycomb visible on the background that could have been mistaken for a bee or a portion of a bee (see the examples in Figure 6). Furthermore, the PPV of class “bee” is just 17.6% with an FDR of 82.4%, meaning that the model often produces “bee” outputs when there are not any. Similarly, the FDR of “blurred_bee” is quite high (30%) and it is mostly due to the incorrect predictions occurring on “background” images (true negative images with no bees in them).

### 4.2. “Bottom Detector” Model Results

Looking at Figure 10, which shows the correlogram plot of all the ground-truth labels of the dataset, we can observe the following:The boxes’ center is located everywhere in the image, more frequently close to $C_{X}=0.8∼1$ and $C_{Y}=0.5∼0.6$ (graph at row 2 column 1). This is also highlighted by the two histograms at (row 1, column 1) and (row 2, column 2).The relationship between $C_{X}$ and *W* (row 3, column 1), $C_{Y}$ and *W* (row 3, column 2), $C_{X}$ and *H* (row 4, column 1), and $C_{Y}$ and *H* (row 4, column 2) is similar, showing that according to the center coordinate considered the corresponding size of the box is always in the range 0∼0.25. This relates to the size of the box according to its location in the image, suggesting that the boxes tend to be small with respect to the image regardless of their position.The boxes tend to have similar sizes along the two dimensions, overall not higher than 0.3, corresponding to the 30% of the image’s original size along both *X* and *Y* dimensions (row 4, column 3). This is also highlighted by the histograms at (row 3, column 3) and (row 4, column 4).

The performance of the model is summarized in Table 2. Metrics were calculated for each sub-dataset (training, validation, and test) and are presented subdivided per class, including a general “all” class that combines the results from all the individual classes. Looking only at the results obtained on the validation and test sub-datasets, we can observe that the performance metrics are very similar and over 90% for precision, recall, and mAP-50. An evident drop in performance can be seen for mAP-50-95 compared to the training sub-dataset performance for class “occluded_bee” (78% versus 67% and 66% respectively, for validation and testing). This is to be expected since the training sub-dataset is bigger than the other two; additionally, it is worth noting that bees of both classes are quite difficult to detect accurately due to the presence of the metallic grid. Nonetheless, the model confidently predicts the “bee” class (84% and 85% mAP-50-95 for validation and test, respectively) despite the instances of this class being fewer (190 boxes of class “bee” versus 610 boxes of class “occluded_bee” among all the 147 images of the test sub-dataset). This result highlights that separating between types of bees is a successful approach, despite using a single “generic bee” class that could have been better in terms of the detection of bees with respect to the background. Moreover, even if the presence of the metallic grid reduces the detection accuracy, the model is still successful in monitoring the bees’ activity.

The confusion matrix of the test sub-dataset is presented in Figure 11, considering that, among the 147 images of the test sub-dataset, a total of 190 boxes of class “bee” and 610 boxes of class “occluded_bee” were present (unbalanced class problem). There were no instances of the “background” class (corresponding to a “no class” or a TN). Data presented in the cells of the matrix have been normalized over the total number of occurrences in the dataset, while TPR, FNR, PPV, and FDR have been obtained as the row-wise and column-wise summary of instances, normalized per class. The TPR for both the “bee” and “occluded_bee” classes exceeds 90%, highlighting the model’s strong ability to accurately distinguish these classes without confusion. However, a significant issue arises with the model’s tendency to misclassify these instances as “background”, resulting in a 100% FNR and FDR for this category. Notably, the PPV is highest for the “bee” class, whereas the “occluded_bee” class experiences more misclassifications, with a 10.2% FDR, primarily due to confusion with the “background” class. Despite these challenges, the model demonstrates remarkable robustness in detecting bees, even in the presence of a metallic grid.

### 4.3. Varroa Destructor

It is worth noting that inside the “Frame” dataset, a few examples of potential *Varroa destructor* mites were spotted (see Figure 12). This finding allows us to conclude two things:The “Frame” inspection point has the potential to help beekeepers identify an ongoing infection and plan their treatments in time.Since in this set-up, we adopted a low-cost camera (as described in Section 2.1), it is evident that its specifications are sufficient for detecting *Varroa destructor* mites with sufficient accuracy. In fact, considering the camera’s specifications (sensor size $SS_{X}$ and $SS_{Y}$ obtained from the datasheet, resolution $R_{X}=R_{Y}=640$ px obtained after resizing the original image) and an average distance of the infected bees from the camera of D=15 mm, we can apply the following formula to find how many pixels correspond to the parasite size:
(1)$$\begin{array}{cc} \Phi_{X} & =\frac{\phi_{X}\cdot f\cdot R_{X}}{D\cdot SS_{X}} \end{array}$$
(2)$$\begin{array}{cc} \Phi_{Y} & =\frac{\phi_{Y}\cdot f\cdot R_{Y}}{D\cdot SS_{Y}} \end{array}$$
for which $\phi_{X}$ and $\phi_{Y}$ are the original width and height of the mite in millimeters (equal to 2 mm for both on average), and *f* is the effective focal length set at 2.6. We obtain that the *Varroa destructor* mite has a diameter $\Phi_{X}$ and $\Phi_{Y}$ equal to 58 px (magnification of 5.7). This value is enough to detect the mite even without the aid of intelligent algorithms. It is worth noting that, since the distance *D* actually changes according to the bees’ movements, the resulting size in pixels may vary. We verified this result by measuring the size in pixels of the spotted mites in the images of Figure 12.

## 5. Discussion

Despite the encouraging results, the presented beehive design and data analysis methodology have some limitations that should be discussed.

First, the data obtained from the “Frame” Camera are mostly blurred due to the movement of the bees and the limited response time of the liquid lenses. A potential solution to this issue is to equip newer lenses and perform a calibration experiment, in which the ideal actuation voltage range is identified according to the bees’ movements. Another idea is to actuate the red illumination to simulate pulsed light, a workaround often used in industrial vision to “freeze” objects in the images despite their fast movements. However, pulsed light could produce undesired stress on the colony which should be properly addressed in a dedicated experiment. Nonetheless, by improving the amount of in-focus images acquired, we can optimize disk space usage and energy consumption of the embedded device. Moreover, the methodology based on the focus measure operators described in Section 3.1 should be modified accordingly (e.g., parameters should be re-tuned to find a new optimal threshold). On a brighter note, it is worth noting that, during the experimentation period, the colony did not attack the camera, leaving it intact, as it was brand new. This suggests that this inspection point bears great potential and should be thoroughly studied in the future to optimize its performance, especially for the detection of *Varroa destructor* mites.

Second, the “Bottom” Camera has a limited view due to the presence of the grid and the distance from the internal frames, reducing its potential as an observation point for the colony’s health. To address this issue, the design of the “Bottom” set-up should be modified to allow the camera to be closer to the frames, either with different optics (e.g., focal length of 12 mm or more to obtain a closer view) or by changing the camera to a smaller one so that it could be placed on top of the grid instead of below it. The potential drawback, in this case, is the cost of the new optics and the availability of the new smaller camera, which will probably have higher signal-to-ratio noise and produce images with optical defects at the corners, both issues due to its sensor size. As a result, the new hardware should be properly tested to find the best solution considering the cost, product availability on the market, power consumption, and acquired data quality. This will be another key point addressed in our future developments.

Concerning the *Varroa destructor* analysis, our work is only a first step in addressing this issue based on the limited data observed. The theoretical findings expressed in Section 4.3 should be validated with a proper experiment in the future involving an infected colony to properly verify if our instrumented beehive can detect the mites on the bees’ bodies with suitable accuracy and robustness. Moreover, extensive tests should be conducted spanning a whole year to validate our approach in a long-term monitoring period.

Finally, it is worth noting that a potential third inspection point can be placed inside the *honey super*, on top of the upper section of the beehive. This could be useful to study the bees’ movements in between frames or when they reach food provided by the beekeeper through the cover. To allow the camera to see inside the beehive, it is necessary to substitute the top with a transparent frame (darkness is guaranteed by the top cover). However, due to the distance of the camera from the red LED strips mounted inside, the obtained images are too dark, thus requiring extra illumination around the optics of the camera and a system to avoid light reflections on the transparent surface. This configuration is still in development; however, a preliminary set-up can be seen in Figure 13.

## 6. Conclusions

The presented work focused on the novel design of an instrumented beehive involving two color cameras, namely, “Frame” (inside the beehive), and “Bottom” (at the bottom of it), an approach that was never tested before in real-world scenarios with success. We experimentally found that the “Frame detector”, despite achieving poor results due to the presence of mostly blurred bees (precision, recall, and mAP-50 around 50% and mAP-50-95 of 46% for class “bee”, over 80% and mAP-50-95 of 67% for class “occluded_bee”, results referring to the test sub-dataset), has great potential for the early detection of the presence of *Varroa destructor* mites since they can be spotted on the in-focus bees quite clearly. On the other hand, the second observation point located at the bottom of the beehive was limited by the presence of the metallic grid separating the two boxes of the beehive. However, contrary to our expectations, the detector properly learned to locate and identify bees, achieving outstanding results despite the challenges (precision, recall, and mAP-50 over 90% for both class “bee” and “occluded_bee”, and an mAP-50-95 of 85% and 66% respectively, results referring to the test sub-dataset).

To conclude, our proposal is a first step toward a new frontier of beehive monitoring that perfectly fits the Agrifood 4.0 paradigm. Further developments will be aimed at improving the presented design, including the third observation point at the top. Moreover, the collected data will be used to develop intelligent algorithms for the colony’s health status monitoring purposes in combination with other sensors, for example, following the breakthroughs of ref. [38], in which a color-based segmentation pipeline was developed.

## Figures and Tables

**Figure 1 sensors-24-05270-f001:**
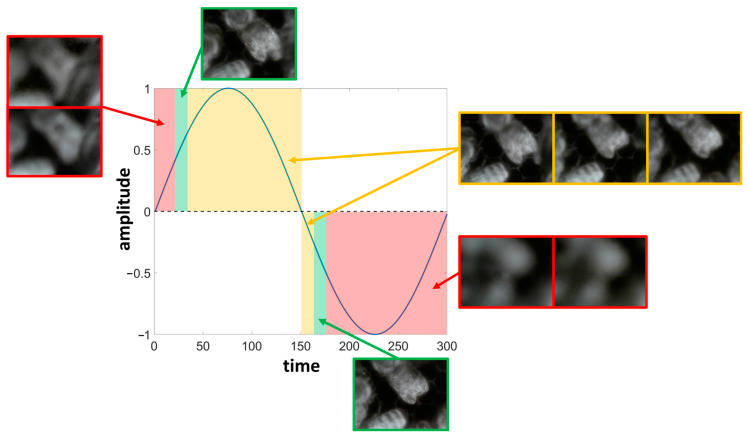
Example of the liquid lenses sweep actuation. Red areas depict focal lengths for which the image is completely out of focus, yellow areas depict focal lengths for which the image is still out of focus but objects are distinguishable from the background, and finally, green areas depict focal lengths for which the output image is in focus.

**Figure 2 sensors-24-05270-f002:**
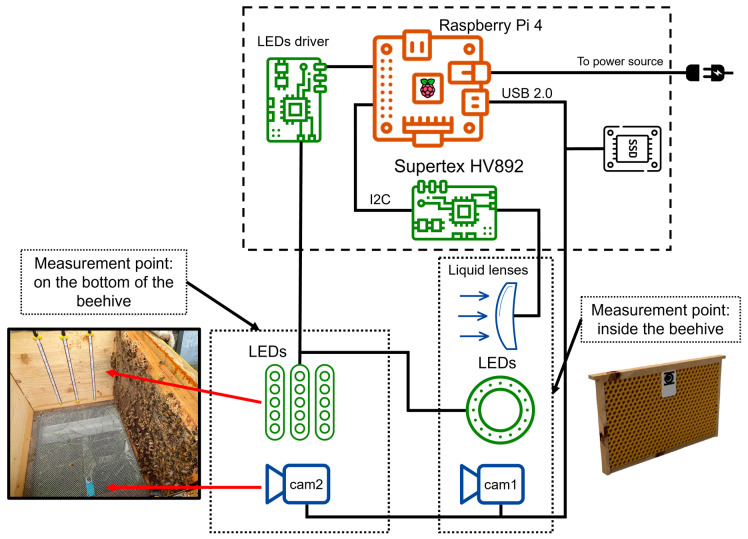
Scheme of the proposed set-up presenting the wiring of electrical components. Images of the mechanical mounting are also shown for both cameras (“Frame” and “Bottom”).

**Figure 3 sensors-24-05270-f003:**
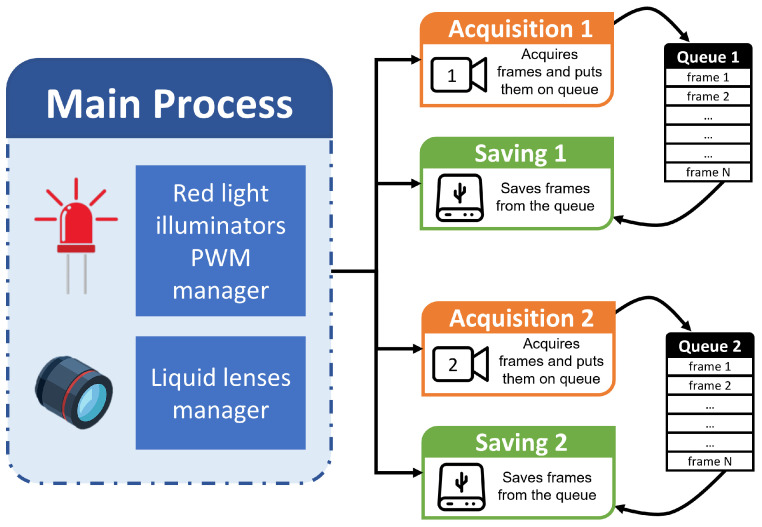
Scheme of the multi-process software that deals with frames acquisition and saving. In this example, only two cameras are shown; however, the software works with up to 4 devices (due to USB bandwidth limitations).

**Figure 4 sensors-24-05270-f004:**
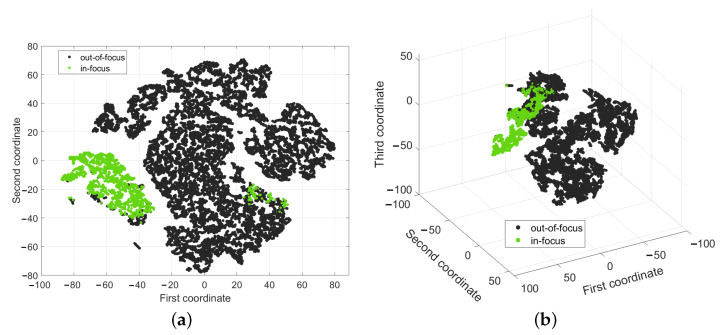
Representation of images subdivision into out-of-focus and in-focus according to the custom thresholding developed. (**a**) The 2D t-SNE visualization. (**b**) The 3D t-SNE visualization.

**Figure 5 sensors-24-05270-f005:**
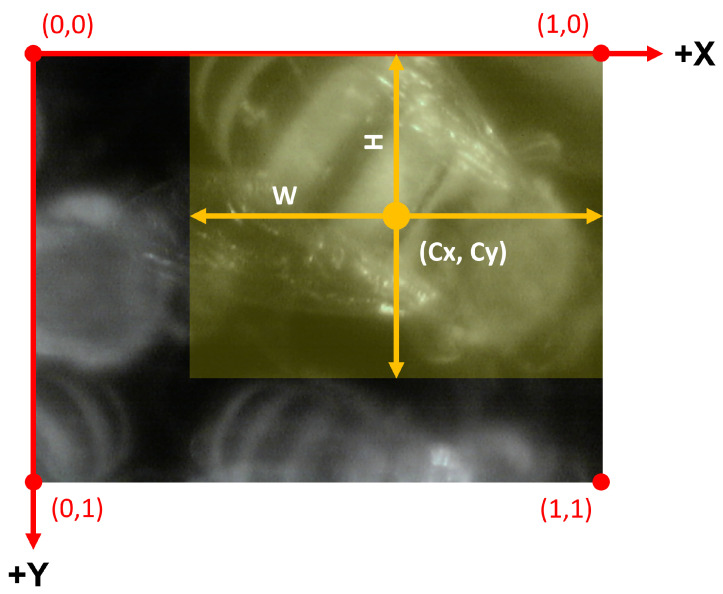
Example of the bounding boxes coordinate system with respect to the image coordinate system. Every parameter is expressed in pixels. The bounding box is depicted in yellow.

**Figure 6 sensors-24-05270-f006:**
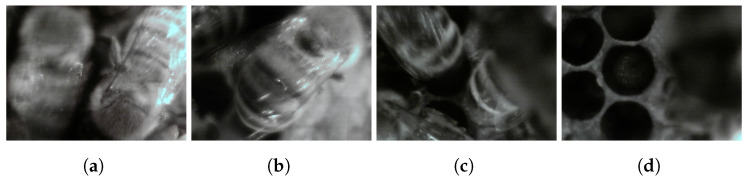
Examples taken from the “Frame detector” dataset. (**a**) The left bee is labeled as “blurred_bee”, and the right bee as “bee” (in-focus). (**b**) Example of a “bee” class fully visible. (**c**) Example of several “blurred_bee” not fully visible. (**d**) Example of a true negative image where no boxes are drawn. A bee or two seem to appear on the right; however, they are so blurred that a confident detection is impossible even for an expert.

**Figure 7 sensors-24-05270-f007:**
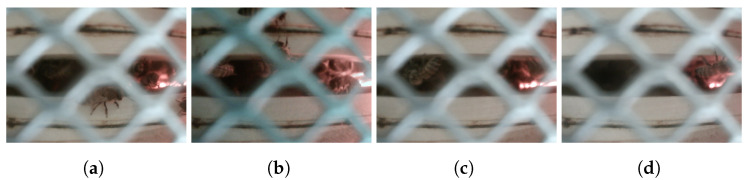
Examples taken from the “Bottom detector” dataset. The visible bees are mostly located on the right (red spot), inside the frames. (**a**) An “occluded_bee” is seen at the bottom center. (**b**) A couple of “occluded_bees” are at the top center. (**c**) Example of a visible “bee” on the left area between the frames. (**d**) Example of a visible “bee” on the right almost outside the frames.

**Figure 8 sensors-24-05270-f008:**
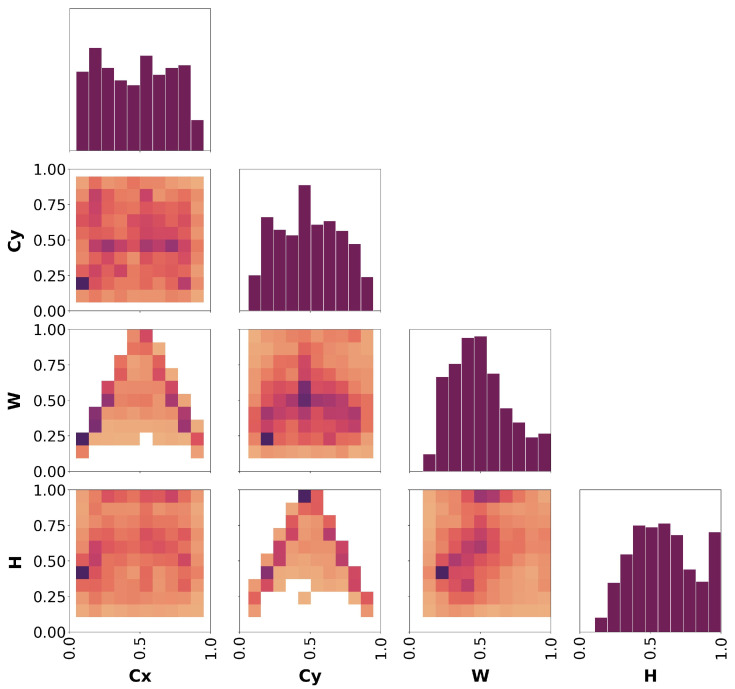
Labels correlogram computed on the “Frame detector” dataset. The diagonal shows the histograms corresponding to the bounding boxes’ coordinates ($C_{X}$, $C_{Y}$, *W*, *H*) respectively. The off-diagonal graphs show the relationship between the four coordinates accordingly.

**Figure 9 sensors-24-05270-f009:**
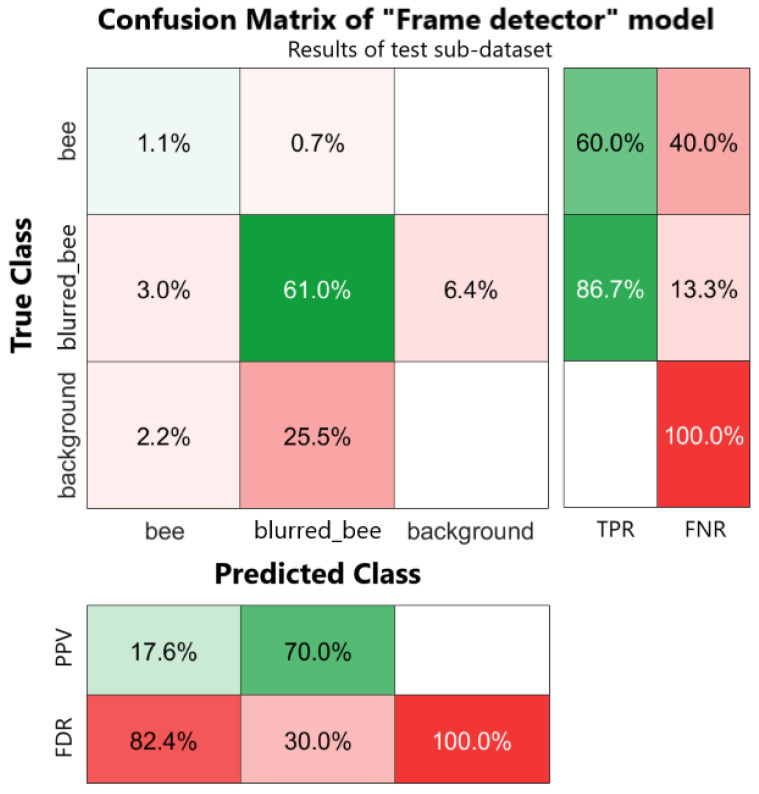
Confusion matrix of the “Frame detector” model computed on resulted data of test sub-dataset. Row-wise and column-wise statistics are shown on the right and at the bottom of the confusion matrix. Values inside the confusion matrix are normalized over the total observations in the test sub-dataset.

**Figure 10 sensors-24-05270-f010:**
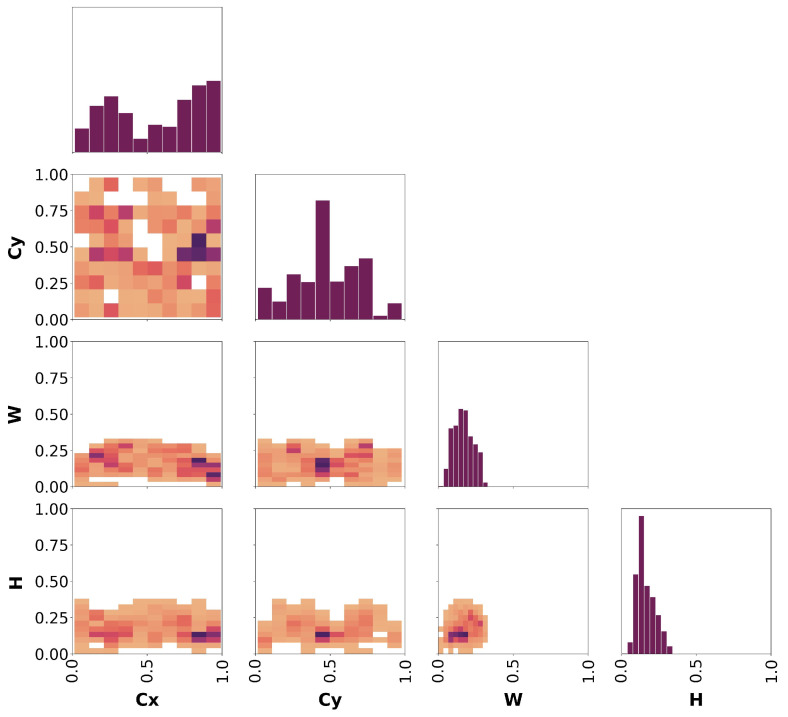
Labels correlogram computed on the “Bottom detector” dataset. The diagonal shows the histograms corresponding to the bounding boxes’ coordinates ($C_{X}$, $C_{Y}$, *W*, *H*) respectively. The off-diagonal graphs show the relationship between the four coordinates accordingly.

**Figure 11 sensors-24-05270-f011:**
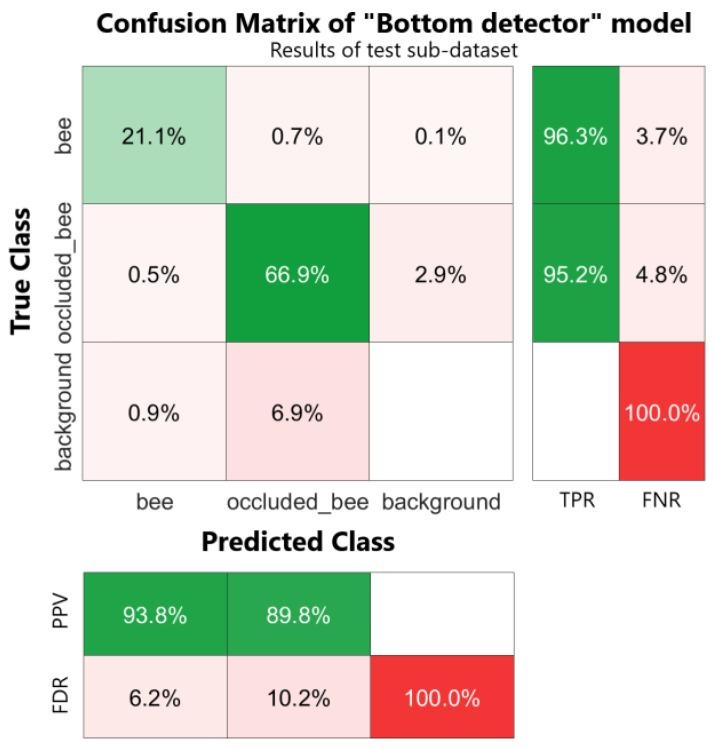
Confusion matrix of the “Bottom detector” model computed on resulted data of test sub-dataset. Row-wise and column-wise statistics are shown on the right and at the bottom of the confusion matrix. Values inside the confusion matrix are normalized over the total observations in the test sub-dataset.

**Figure 12 sensors-24-05270-f012:**
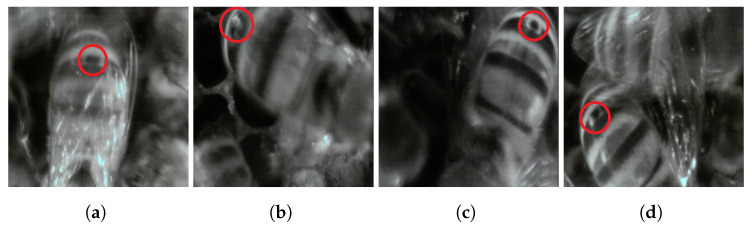
(**a**–**d**) Examples of potential *Varroa destructor* mites taken from the “Frame” dataset highlighted with a red circle on top of the original image.

**Figure 13 sensors-24-05270-f013:**
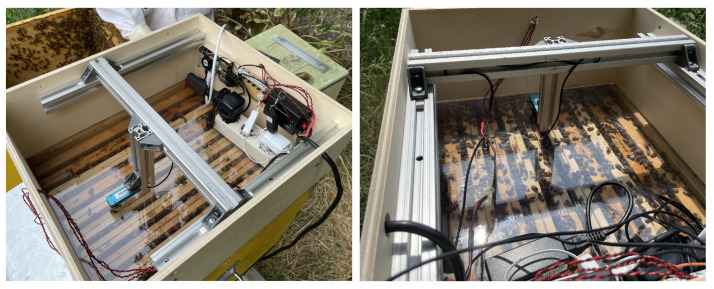
Top view of the set-up depicting the inside of the *honey super*, where the third camera and electronic components are placed. This configuration is still in development.

**Table 1 sensors-24-05270-t001:** Performance metrics of the “Frame detector” model.

Data	Class	Precision	Recall	mAP-50	mAP-50-95
Training	all	76%	79%	85%	71%
	bee	61%	69%	74%	64%
	blurred_bee	90%	89%	95%	78%
Validation	all	60%	79%	70%	53%
	bee	44%	70%	52%	43%
	blurred_bee	77%	88%	88%	64%
Test	all	68%	65%	71%	57%
	bee	50%	50%	53%	46%
	blurred_bee	86%	80%	88%	67%

**Table 2 sensors-24-05270-t002:** Performance metrics of the “Bottom detector” model.

Data	Class	Precision	Recall	mAP-50	mAP-50-95
Training	all	97%	99%	99%	82%
	bee	97%	99%	98%	87%
	occluded_bee	97%	98%	99%	78%
Validation	all	96%	96%	97%	76%
	bee	97%	98%	98%	84%
	occluded_bee	95%	94%	96%	67%
Test	all	94%	96%	96%	75%
	bee	96%	98%	98%	85%
	occluded_bee	92%	94%	94%	66%

## Data Availability

The original data presented in the study are openly available as a public dataset at ref. [39].

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
