# Peer review of "Sensorizing a Beehive: A Study on Potential Embedded Solutions for Internal Contactless Monitoring of Bees Activity"

_sensors, 2024, doi:10.3390/s24165270_

Round 1

Reviewer 1 Report

Comments and Suggestions for Authors

This paper proposes a novel design of an instrumented beehive involving color cameras placed inside the beehive and at the bottom of it. The overall acquisition system is described focusing on design choices towards  an effective solution for precision beehive monitoring. The methods and the design is interesting, and the experiments results validate the performance. The comments are sumamrized as below.

1. Some statistical tests, such as Wilcoxon and ANOVA, should be performed to ensure the quality of the proposed method.

2. There should be some discussion on the limitations of the methods presented in a separate section.

3. Improve the English of the work. Proofreading is recommended.

4. Has the training parameters in constructing the designed model been fine-tuned through preliminary experiments? Also, the concrete software environment parameters such as operating system, programming language and deep learning framework should be given in experiments.

Comments on the Quality of English Language

Proofreading is recommended.

Author Response

Comments 1: This paper proposes a novel design of an instrumented beehive involving color cameras placed inside the beehive and at the bottom of it. The overall acquisition system is described focusing on design choices towards  an effective solution for precision beehive monitoring. The methods and the design is interesting, and the experiments results validate the performance.

Response 1: We thank the reviewer for taking its time to carefully review our work and for his/her words of appreciation.

Comments 2: Some statistical tests, such as Wilcoxon and ANOVA, should be performed to ensure the quality of the proposed method.

Response 2: We thank the reviewer for their concern. It is unclear to us where these statistical tests should be performed, since the methodology presented is subdivided into three topics: (i) the design of the instrumented beehive, including sensors and data acquisition devices; (ii) a procedure to automatically separate in-focus images from out-of-focus ones; and (iii) a qualitative analysis of the inspection points ability to detect bees and their movements according to the results of two object detectors trained on the acquired images. We think that in (ii) there’s space for a t-SNE analysis to demonstrate that the thresholding applied is effective (see Section 3.1 for the modified version); however, there is no need for statistical tests for the other topics. In fact, Wilcoxon, Anova, and similar tests are not suited for image-based analysis since they are more relevant for observations comprehending other types of quantitative measurements, which is not our case. As for topic (iii), it has been extensively analyzed using other statistical tools relevant for the computer vision and deep learning communities (e.g., the correlogram, the confusion matrix, and detection metrics such as precision, recall, and mAP shown in the tables in Section 4). We expanded Section 3.2.3 to include a bit more explanation about these metrics to better clarify their relevance for our work and their statistical significance. Please refer to the corresponding literature cited in the text for more information.

Comments 3: There should be some discussion on the limitations of the methods presented in a separate section.

Response 3: We are grateful for this suggestion. We included Section 5 “Discussion” to address all the limitations of the presented work with potential solutions that we aim to address in the future development of our system. To this end, we moved Fig. 12 and the corresponding discussion (previously in the Conclusions section) to the new Section 5.

Comments 4: Improve the English of the work. Proofreading is recommended.

Response 4: As suggested, we carefully revised the manuscript and corrected all the grammar mistakes.

Comments 5: Has the training parameters in constructing the designed model been fine-tuned through preliminary experiments? Also, the concrete software environment parameters such as operating system, programming language and deep learning framework should be given in experiments. 

Response 5: The model used was YOLO v8 developed by Ultralytics. The library gives access to a pre-trained version of the model, which we used to perform transfer learning and obtain our trained models quickly despite the limited amount of training data. The training platform was Google Colab, which is a web service allowing users to train models on Google machines equipped with GPUs. We used the standard plan, which includes a machine with 12 GB RAM, disk space of 78 GB and a GPU of 15 GB. All the training and testing code was developed in Python 3. We included this information in Section 3.2.

Reviewer 2 Report

Comments and Suggestions for Authors

Manuscript title: Sensorizing a Beehive: A Study on Potential Embedded Solutions for Internal Contactless Monitoring of Bees Activity

Overall, the manuscript contains some useful information worthy of publication. However, major revisions are needed.

Comments

  1. The authors should emphasize the novel aspects of their bee monitoring approach. Additionally, they should explain how this approach could create a new monitoring paradigm for beekeepers.
  2. The authors should address the potential influence of the two red light illuminators on bee behaviour.
  3. From a statistical perspective, the authors must justify whether the total number of good images (2057) is appropriate.
  4. The text in lines 246-256 contains repetition and needs to be revised for clarity and conciseness.
  5. The text in lines 275-284 requires revision to present the rationale and justification for the approach.
  6. The statement in lines 294-295 assumes no labelling errors, which could be a major threat to the validity of the results. This issue should be addressed.
  7. Stronger statements are needed in lines 413-420 to emphasize key findings and their significance.
  8. The text in lines 441-451 summarizes what has been done and is repetitive. This text should be removed, and the focus should shift to key outcomes in the conclusion section.

The limitations of the proposed design should be clearly stated in the conclusion section.

Comments on the Quality of English Language

Moderate editing of the English language required

Author Response

Comments 1: Overall, the manuscript contains some useful information worthy of publication. However, major revisions are needed.

Response 1: We thank the reviewer for their time revising our work. We addressed their concerns and modified the submitted manuscript accordingly. The following is a point-by-point response to their inquiries.

Comments 2: The authors should emphasize the novel aspects of their bee monitoring approach. Additionally, they should explain how this approach could create a new monitoring paradigm for beekeepers.

Response 2: We thank the reviewer for this suggestion. We modified the final portion of the Introduction Section to highlight this aspect better. Moreover, the modified version of the Abstract and Conclusions Sections also stresses the novelty and advances of our proposed design.

Comments 3: The authors should address the potential influence of the two red light illuminators on bee behaviour.

Response 3: We are grateful for this comment. We included this discussion in Section 2.1 “Developed set-up” backed up with a literature reference. In summary, red light has been proven by the scientific community to have positive effects on bee colonies, especially during winter, since it’s the least disturbing wavelength for them and also aids them in maintaining heat inside the colony during adverse weather conditions (e.g., snow, rain, sudden temperature drops), as well as improving mitochondrial activity to defend them against the effects of pesticides.

Comments 4: From a statistical perspective, the authors must justify whether the total number of good images (2057) is appropriate.

Response 4: We understand the reviewer's concern. However, in computer vision, this is typically unnecessary because it depends on the model and the training procedure adopted. In our case, we chose an off-the-shelf YOLO v8 model that was pre-trained by Ultralytics (the developers) on large image datasets; hence, the model was already mostly tuned to perform object detection with high accuracy levels. We re-trained the model following a transfer learning procedure, so only a few hundred images are sufficient to reach satisfactory results. In general, transfer learning is used when the type of data is the same between the pre-trained model A and the desired model B (e.g., images), but their content is different (e.g., A is trained to recognize cats, B is trained to recognize ships). We added a few lines in Section 3.2 and a reference about transfer learning. To answer the reviewer's question, yes, more than 1000 images are enough for transfer learning to succeed. By looking at the confusion matrix reported in the Results Section, it is possible to check the performance of the trained model on the Test dataset. The only concern regards the class balancing (e.g., the number of occurrences of class “bee” with respect to class “blurred_bee”). They are unfortunately unbalanced, and this is the reason why the metrics in Table 1 are around 50% for class “bee” compared to the over 80% values of class “blurred_bee”. However, this is by no means a consequence of a bad methodology (either for acquisition or processing of the images), but a consequence of the phenomenon studied (e.g., the movements of the bees inside the beehive) and the short period of the experiment. The discussion about these limitations and potential methods to mitigate them in future developments is presented in Section 4.1 and Section 5.

Comments 5: The text in lines 246-256 contains repetition and needs to be revised for clarity and conciseness.

Response 5: We thank the reviewer for noting this. After a careful revision, we decided to keep lines 246-256 about the “Frame” data reduction, but we removed the repetition happening at lines 226-228 of the previous submission.

Comments 6: The text in lines 275-284 requires revision to present the rationale and justification for the approach.

Response 6: Concerning the lines highlighted by the reviewer, we want to stress that the subdivision presented is commonly used by the research community dealing with machine learning and deep learning models. The validation sub-dataset aims to evaluate the capability of the chosen model to avoid overfitting and the overall performance of the trained model according to the hyperparameters learned. If the performance on the validation sub-dataset is unsatisfactory, the model should be re-trained. The test sub-dataset, on the other hand, tests the trained model on unseen data. We added a reference and a brief explanation to support our decision in Section 3.2.1.

Comments 7: The statement in lines 294-295 assumes no labelling errors, which could be a major threat to the validity of the results. This issue should be addressed.

Response 7: As the reviewer noted, the only way to be sure that no labeling errors occurred in the dataset is to manually check them with experts. Hence, we double-checked our labels with another expert in the field and found no mistakes. To avoid confusion, we modified the corresponding statement in Section 3.2.3.

Comments 8: Stronger statements are needed in lines 413-420 to emphasize key findings and their significance.

Response 8: As suggested by the reviewer, we rephrased the paragraph mentioned (Section 4.2, just before Section 4.3) following their instructions.

Comments 9: The text in lines 441-451 summarizes what has been done and is repetitive. This text should be removed, and the focus should shift to key outcomes in the conclusion section.

Response 9: Following the reviewer's suggestion, we modified the Conclusions paragraph and removed repetitions.

Comments 10: The limitations of the proposed design should be clearly stated in the conclusion section.

Response 10: We thank the reviewer for this comment. We added a new section called “Discussion” in which we moved all the discussion related to the drawbacks, limitations, and potential improvements of the set-up and methodology to be addressed in future developments.

Reviewer 3 Report

Comments and Suggestions for Authors

Thank you for the nice work. This research topic is interesting and informative to readers. Please remove the number alignment on pages 6, 9, 10, 12, 14 respectively. You could use bullets or Roman numbers. 

Please add a discussion part separately as 4.4. 

Please add the 'Data Availability Statement', and 'Institutional Review Board Statement' if applicable. 

Author Response

Comments 1: Thank you for the nice work. This research topic is interesting and informative to readers. Please remove the number alignment on pages 6, 9, 10, 12, 14 respectively. You could use bullets or Roman numbers. 

Response 1: First, we wish to thank the reviewer for taking the time to review the manuscript and showing appreciation for our work. Following their requests, we changed the numerical bullets to point bullets.

Comments 2: Please add a discussion part separately as 4.4.

Response 2: We thank the reviewer for this comment. We added a new section called “Discussion” to also address the limitations of the set-up. Discussion about the results (related to the detection performance) was kept inside the Results section because we think it’s the best way to help readers understand the paper's conclusions.

Comments 3: Please add the 'Data Availability Statement', and 'Institutional Review Board Statement' if applicable.

Response 3: As suggested by the reviewer, we added the 'Data Availability Statement' including all our data as a Mendeley Data downloadable dataset. As for the 'Institutional Review Board Statement', we chose not to include it because, at the present time, insects are not classified as animals for the Italian Ethical Committees so we do not require an Ethical Committee approval to conduct research on them. 

Reviewer 4 Report

Comments and Suggestions for Authors

As bees can see the ultraviolet spectrum, it is possible that there are structures at the entrance to the hive (covered by sunlight with its ultraviolet composition), I suggest the use of multispectral cameras for future work on this research, in order to obtain more information to correlate behavior with lights. In any case, the project was well done and technically sophisticated in its implementation.

Author Response

Comments 1: As bees can see the ultraviolet spectrum, it is possible that there are structures at the entrance to the hive (covered by sunlight with its ultraviolet composition), I suggest the use of multispectral cameras for future work on this research, in order to obtain more information to correlate behavior with lights. In any case, the project was well done and technically sophisticated in its implementation.

Answer 1: We thank the reviewer for this kind suggestion! We will surely experiment with UV light and other light sources to further investigate bees' behavior.

Round 2

Reviewer 2 Report

Comments and Suggestions for Authors

The revised version of the manuscript is a much better version than the original submission. I have no further comments.

Comments on the Quality of English Language

Moderate editing of the English language is required.